# DNA Polymerase Theta Plays a Critical Role in Pancreatic Cancer Development and Metastasis

**DOI:** 10.3390/cancers14174077

**Published:** 2022-08-23

**Authors:** Agnieszka Smolinska, Kerstin Singer, Janine Golchert, Urszula Smyczynska, Wojciech Fendler, Matthias Sendler, Jens van den Brandt, Stephan Singer, Georg Homuth, Markus M. Lerch, Patryk Moskwa

**Affiliations:** 1Department of Medicine A, University Medicine Greifswald, 17475 Greifswald, Germany; 2Institute of Pathology, University Medicine Greifswald, 17475 Greifswald, Germany; 3Department of Pathology and Neuropathology, University Hospital Tuebingen, 72076 Tuebingen, Germany; 4Department of Functional Genomics, Interfaculty Institute for Genetics and Functional Genomics, University Medicine Greifswald, 17475 Greifswald, Germany; 5Department of Biostatistics and Translational Medicine, Medical University of Lodz, 92-215 Lodz, Poland; 6Department of Radiation Oncology, Dana-Farber Cancer Institute, Boston, MA 02215, USA; 7Central Core and Research Facility of Laboratory Animals (ZSFV), University Medicine Greifswald, 17475 Greifswald, Germany; 8Ludwig Maximilian University Hospital, Ludwig Maximilian University of Munich, 81377 Munich, Germany

**Keywords:** PDAC, DNA repair, non-homologous end joining, polymerase theta

## Abstract

**Simple Summary:**

Pancreatic ductal adenocarcinoma (PDAC) is one of the most deadly cancers worldwide. The occurrence of oncogenic KRAS mutations is considered a signature event in PDAC, leading to genomic instability. The aim of our study was to evaluate the impact of the oncogenic KRAS G12D mutation on the activity of the error-prone alt-EJ repair mechanism, and to investigate the potential role of Polθ in the development of pancreatic cancer. We found that oncogenic KRAS increases the expression of key alt-EJ proteins in a mouse and human PDAC model. Using TLR assay, we also found increased alt-EJ activity in mouse and human cell lines upon the expression of KRAS D12D. The inactivation/impairment of alt-EJ by polymerase theta (Polθ) depletion delays the development of pancreatic cancer and prolongs the survival of experimental mice, though it does not prevent the PDAC development, which leads to full-blown PDAC with disseminated metastasis. Our studies provide a high-value target as a novel therapeutic candidate for the treatment of pancreatic and other cancers.

**Abstract:**

Pancreatic ductal adenocarcinoma (PDAC), due to its genomic heterogeneity and lack of effective treatment, despite decades of intensive research, will become the second leading cause of cancer-related deaths by 2030. Step-wise acquisition of mutations, due to genomic instability, is considered to drive the development of PDAC; the KRAS mutation occurs in 95 to 100% of human PDAC, and is already detectable in early premalignant lesions designated as pancreatic intraepithelial neoplasia (PanIN). This mutation is possibly the key event leading to genomic instability and PDAC development. Our study aimed to investigate the role of the error-prone DNA double-strand breaks (DSBs) repair pathway, alt-EJ, in the presence of the KRAS G12D mutation in pancreatic cancer development. Our findings show that oncogenic KRAS contributes to increasing the expression of Polθ, Lig3, and Mre11, key components of alt-EJ in both mouse and human PDAC models. We further confirm increased catalytic activity of alt-EJ in a mouse and human model of PDAC bearing the KRAS G12D mutation. Subsequently, we focused on estimating the impact of alt-EJ inactivation by polymerase theta (Polθ) deletion on pancreatic cancer development, and survival in genetically engineered mouse models (GEMMs) and cancer patients. Here, we show that even though Polθ deficiency does not fully prevent the development of pancreatic cancer, it significantly delays the onset of PanIN formation, prolongs the overall survival of experimental mice, and correlates with the overall survival of pancreatic cancer patients in the TCGA database. Our study clearly demonstrates the role of alt-EJ in the development of PDAC, and alt-EJ may be an attractive therapeutic target for pancreatic cancer patients.

## 1. Introduction

Pancreatic ductal adenocarcinoma, due to its genomic heterogeneity and the lack of development of an efficient treatment, is expected to become the second leading cause of cancer-related deaths by 2030 [1]. Pancreatic cancer is not a de novo disease, but may arise from the low- and high-grade premalignant lesions designated as pancreatic intraepithelial neoplasia (PanINs) that ultimately progress to invasive cancer [2]. These lesions already carry a very high penetrance oncogenic KRAS mutation, and the most prevalent mutation in two thirds of cases is G12D [3]. Other variants are G12C and G12R; all of these result in the activation of KRAS by reducing the hydrolysis of GTP-bound KRAS [4]. All three mutations differ from each other in impact on premalignant lesion formation, tumor initiation, and response to EGFR inhibitors [5]. Regarding the oncogenic potential of G12C and G12R, there is some controversy [6]. As in the original papers by Hingorani et al., the most frequent mutation G12D was used in the animal models and pancreatic cell lines [7,8]. The occurrence of this mutation may be the initial step in the tumorigenesis of pancreatic cancer, leading to the genomic instability indispensable for the sequential inactivation of suppressor genes such as p53, p16 (CDKN2A/INK4A), and DPC4 (SMAD4/DPC4) [9]. The mechanism of genomic instability induced by oncogenic KRAS, and the role of DNA repair in this process, is poorly understood [8].

A newly emerging entity of genomic instability that is linked to oncogenic KRAS is the alternative, non-homologous end-joining (alt-EJ), also known as microhomology-mediated end joining (MMEJ), a poorly understood DNA double-strand break (DNA DSB) repair mechanism. Although alt-EJ was initially considered as a backup DNA repair pathway, recent studies show that alt-EJ also functions in the presence of canonical, non-homologous end joining (c-NHEJ) and homologous recombination (HR), which confirms that it may be the only available repair pathway for specific types of DNA damage [10,11,12,13,14,15,16]. Repair by alt-EJ is driven by the annealing of micro-homologous sequences flanking the DNA ends, and its outcome is mutagenic due to deletions and insertions that scar break sites. Due to inappropriate repair, alt-EJ may also promote tumorigenesis by increasing genomic instability.

According to current studies, the best-established components linked to alt-NHEJ are poly(ADP-ribose) polymerase 1 (PARP1), DNA ligase 3 (LIG3), DNA ligase 1 (LIG1), Xrcc1, CtIP, Mre11, and polymerase theta (Polθ), also known as polymerase Q (POLQ) [17].

Microhomology-mediated end joining mutagenicity is attributed to the promiscuous activity of Polθ, a unique polymerase, as it contains a helicase-like domain at its N-terminus, in addition to a polymerase domain at its C-terminus [18]. Polθ is suppressed in normal human tissue and, in contrast, is upregulated in a wide range of cancers, including lung, gastric, and colorectal [19]. In addition, a high level of Polθ is present in HR-deficient cancers that rely on its backup activity for survival [20]. Moreover, patients with high levels of Polθ expression have a significantly poorer clinical outcome compared with those expressing low levels of Polθ [20,21,22]. Notably, there is growing evidence that polymerase theta may play a key role in DNA repair, and its investigation may reveal new therapeutic targets for cancer treatment. Therefore, the knockout of polymerase theta, a crucial component of the error-prone pathway in pancreatic cancer, may either prevent or delay its development due to reduced mutability and, consequently, extend overall survival.

## 2. Materials and Methods

### 2.1. Cell Lines

The mouse pancreatic cancer cell line Panc02 was provided by Tuveson Laboratory (Cold Spring Harbor Laboratory Cancer Center). The BxPC3, human pancreatic cancer cell line was a generous gift from Dr. Giese (University Hospital Heidelberg). Variants of Panc02 and BxPC3 cell lines with exogenous KRAS wild-type expression or oncogenic KRAS carrying the G12D mutation were generated by the lentiviral transduction system. Cells were cultured in DMEM or RPMI, and supplemented with 10% FBS, penicillin, and streptomycin at 37 °C, and 5% CO_2_.

### 2.2. DNA Sequencing

Genomic DNA from cells was extracted using PureLink™ Genomic DNA Mini Kit (Thermo Fisher, Waltham, MA, USA), according to the manufacturer’s protocol. The isolated gDNA was further used for PCR amplification of the target sequence, followed by standard protocol. After PCR reaction, amplicons were purified using AMPure XP beads (Beckman Coulter, Brea, CA, USA) and then prepared for II PCR sequencing. In this step, the BigDye Terminator v3.1 kit (Thermo Fisher, Waltham, MA, USA) was used following the manufacturer’s protocol. Analysis of mutation of the genes of interest (KRAS, Trp53) was performed using the Applied Biosystems 3130xl Genetic Analyzer (Applied Biosystems, Waltham, MA, USA).

### 2.3. Lentiviral Production and Infection

HEK-293T cells were co-transfected with lentiviral packaging plasmids (pCMV-dR8.91, pMD2.G-VSVG) and transfer plasmids (pLVXDsRed-Monomer-C1 expressing KrasWT or KrasG12D) by calcium phosphate precipitation method (CalPhos™ Mammalian Transfection Kit, Takara Bio 631312, Mountain View, CA, USA), following the manufacturer’s protocol. The supernatant containing the virus was collected 48 h later and concentrated using PEG Virus Precipitation Kit (BioCat K904-50/200, Heidelberg, Germany). Cells were transduced with lentivirus particles in the presence of polybrene. For selection of stably infected cells, 1.5–2 µg/mL of puromycin was added. Transduction efficiency was checked on a BD LSR II flow cytometer (BD Biosciences, San Jose, CA, USA).

### 2.4. Microarray Analysis

Total RNA was isolated from cells using the RNeasy Mini Kit (Qiagen 74106, Hilden, Germany), according to the manufacturer’s protocol. Obtained RNA samples were purified using the RNA Clean-Up and Concentration Micro Kit, and quality was checked by the Agilent 2100 Bioanalyzer. For further microarray analysis, RNA samples with an RNA integrity number (RIN) ≥ 9.0 were used. The microarray analysis was carried out using individual RNA samples (n = 3) that were processed following the manufacturer’s instructions of the GeneChip^TM^ WT PLUS Reagent Kit, and hybridized with GeneChip™ Mouse Gene 2.0 ST Assay or GeneChip™ Human Gene 2.0 ST Assay. The quality control of hybridizations and data analysis were conducted in Transcriptome Analysis Console. All data were normalized using a robust multi-array average (RMA) algorithm. The microarray data analysis was performed using the R/Bioconductor package oligo and Rosetta Resolver software system. To identify significantly differentially expressed genes (*p* < 0.05, fold change ≥ 1.5-fold), one-way ANOVA and *t*-tests were performed. Significantly differentially expressed genes and common crucial pathways between experimental groups were subsequently identified by Ingenuity Pathway Analysis (IPA) and Gene Set Enrichment Analysis (GSEA) [23].

### 2.5. Quantitative PCR (qPCR)

Total RNA was prepared with the RNeasy Mini Kit (Qiagen 74106, Hilden, Germany), and transcribed using random hexamers and MMLV reverse transcriptase (Epicentre TR80125K). The quantitative expression of the mRNA was measured with QuantStudio 7 Flex real-time PCR (Life Technologies, Carlsbad, CA, USA) using the SYBR Select Master Mix (Applied Biosystems, Waltham, MA, USA), according to the manufacturer’s manual. The relative expression of studied genes was normalized to the expression of reference gene 5S rRNA.

### 2.6. Western Blotting

A total lysate of 30 μg protein was loaded on SDS-Gels and transferred to nitrocellulose membranes. The membranes were incubated with primary antibodies overnight and, subsequently, probed with secondary HRP-conjugated antibodies. Primary antibodies: GAPDH (Meridian Bioscience H86504M, Cincinnati, OH, USA), Ku70 (Santa Cruz Biotechnology sc-1486, Heidelberg, Germany), Ku80 (Santa Cruz Biotechnology sc-9034, Heidelberg, Germany), DNA ligase IV (Santa Cruz Biotechnology sc-28232, Heidelberg, Germany), Mre11 (Cell Signaling 4895, Danvers, MA, USA), DNA ligase III (BD Transduction Laboratories 611876, Franklin Lakes, NJ, USA), PARP (Cell Signaling 9542), DNA polymerase theta (Abcam ab80906, Cambridge, UK).

### 2.7. Cell Cycle Analysis

Panc02 cells and BxPC3 cells either expressing KrasWT or KrasG12D were fixed in 70% ethanol. Subsequently, the cells were analyzed in PI/RNase staining buffer (BD Pharmingen 550825, San Diego, CA, USA) with the BD LSR II flow cytometer (BD Biosciences, Allschwil, Switzerland).

### 2.8. MTT Assay

To obtain growth curves of the generated variants of Panc02 and BxPC3 cells, 3.5 × 10^3^ of Panc02 and 5 × 10^3^ of BxPC3 cells were seeded in a 96-well plate containing 100 µL of DMEM or RPMI medium, respectively. Cells were grown up to different time points: 24 h, 48 h, and 72 h. On the day of measurement, 10 µL of MTT was added to the cells and incubated for 3 h. After the incubation time, the insoluble formazan was solubilized with 150 µL of acidic isopropanol (0.04 M HCl in absolute isopropanol). The quantity of formazan was measured at 570 nm in SpectraMax Plus 384 Microplate Reader. All assays were performed three times independently.

### 2.9. The Traffic Light Reporter

To measure the mutagenic alt-NHEJ activity in Panc02 and BxPC3 cell lines, a TLR reporter system was used. The pCVL Traffic Light Reporter 1.1 Ef1a Puro plasmid (pCVL-TLR), together with lentiviral packaging plasmids (pCMV-dR8.91 and pMD2.G-VSVG), was introduced into HEK293 LentiX cells using the calcium phosphate precipitation method (CalPhos™ Mammalian Transfection Kit, Takara 631312), according to the manufacturer’s protocol. Afterwards, 4 × 10^4^ of Panc02 and 1 × 10^5^ of BxPC3 cell variants were seeded in a 12-well plate and infected with the prepared viral plasmid (pCVL-TLR). Cells were transduced with 5-fold serial dilutions of the lentiviral stocks, followed by puro selection to estimate the lentivirus titer. Subsequently, Panc02 and BxPC3 cell lines containing single virus particles with TLR construct were infected with the pCVL SFFV GFP EF1s HA NLS Sce (opt) viral plasmid (donor with I-SceI endonuclease), using lentiviral transduction. Production of I-SceI lentiviral particles was performed using the calcium phosphate method. The activity of mutNHEJ (mCherry) and HR (GFP) was measured by flow cytometry on BD LSR II flow cytometer (BD Biosciences).

### 2.10. Animal Models

Mice were housed in standard specific-pathogen-free conditions in the Central Core and research facility of Laboratory Animals at the University Medicine Greifswald. All experiments were performed and approved according to the regulations of Greifswald University. For the animal studies, p48^+/Cre^; LSL-Kras^G12D/+^ (KC) mice and p48^+/Cre^; LSL-Kras^G12D/+^; Polq^tm1Jcs^ (qKC) mice were used. In the KC mouse model, previously described by Hingorani et al. [7], LSLKras^G12D/+^ animals were bred with p48^+/Cre^ animals. Coexistence of p48Cre and Kras^G12D^ locus results in expression of Cre recombinase and consecutive expression of KrasG12D mutation. To generate the qKC mouse model, LSL-Kras^G12D/+^ and p48^+/Cre^ animals were crossed on POLQ deficient background using Polq^tm1Jcs^ mice purchased from Jackson Laboratory. Progeny from these groups was further cross-bred to produce p48^+/Cre^; LSL-Kras^G12D/+^; Polq^tm1Jcs^ (qKC) mice.

### 2.11. Patient Sample Collection

All pancreatic tissue were collected from patients who underwent pancreatic surgery due to PDAC. The tumor was confirmed by histological analysis performed by a pathologist at the Department of Pathology, University Medicine Greifswald. Healthy tissues were obtained from the healthy edge surrounding the tumor. The patients did not undergo any chemotherapeutic or radiation treatment before surgery. All tissues were collected according to the protocol set by the ethics committee.

### 2.12. Histology, Immunohistochemistry, and Alcian Blue

H&E staining, immunohistochemistry, and alcian blue staining were performed from paraffin-embedded tissue samples; 1–2 µm slides were cut by microtome (Leica). All antibodies for immunohistochemical staining were used in 1:50, 1:100, or 1:200 dilution, and incubated overnight at 4 °C. Primary antibodies: DNA polymerase theta (Abcam ab111218, Cambridge, UK), PARP1 (Abcam ab32138, Cambridge, UK), Mre11 (Cell Signaling 4895, Danvers, MA, USA), Ku70 (Santa Cruz Biotechnology sc-1486, Heidelberg, Germany), Ki67 (Bethyl Laboratories IHC-00375, Montgomery, TX, USA), PCNA (Cell Signaling 2586, Danvers, MA, USA), CyclinD1 (Cell Signaling 2978, Danvers, MA, USA), p-ERK1 (Cell Signaling 4370, Danvers, MA, USA), Cox2 (Cell Signaling 12282, Danvers, MA, USA). Alcian blue was performed by Alcian Blue pH 2.5 Stain Kit (Vector Laboratories H-3501, Newark, CA, USA), according to the manufacturer’s protocol. All slides used for histological analysis were scanned with the Pannoramic Midi II (3DHISTECH, Budapest, Hungary), and evaluated with different magnification using Quant Center software (3DHISTECH, Budapest, Hungary).

### 2.13. Statistical Analysis

Statistical analysis was carried out with GraphPad Prism software v.9.4.1 (San Diego, CA, USA). Data from in vivo and in vitro experiments were plotted either with the mean value plus standard deviation (SD), or standard error of the mean (SEM). Significant differences were analyzed by unpaired student *t*-test to compare between two variables, and one-way analysis of variance (ANOVA) for multiple comparisons. Survival curves were analyzed with the Mantel–Cox test. A variance with a *p*-value < 0.05 was considered significant.

## 3. Results

### 3.1. Oncogenic KRAS^G12D^ and Its Impact on the Protein Expression of the Alt-EJ Repair Pathway

KRAS is the most mutated oncogene in human cancers, with the highest frequency in pancreatic cancer (about 100%) [24]. Numerous studies report that mutations of the KRAS gene play an important role in PDAC development [25]. Furthermore, the activation of oncogenic KRAS can affect DNA repair pathways, causing abnormal repair and accumulation of genomic alterations. Therefore, we first investigated whether the mutagenic KRAS affects the alt-EJ repair pathway in pancreatic cancer cell lines. For this purpose, we used murine Panc02 and human pancreatic cancer cell line BxPC3. Both of these cell lines do not harbor activating KRAS mutations. To confirm the absence of the KRAS G12D mutation, the most common mutation in PDAC, we employed the Sanger sequencing in the Panc02 and BxPC3 cell lines. In addition, we checked for the presence of another mutated gene, TP53, the second most frequently occurring in pancreatic cancer and associated with KRAS activation. TP53 encodes a tumor suppressor transcription factor, p53, which mediates many antiproliferative effects in response to a variety of stress factors, including DNA damage. Most known mutations are in the DNA binding domain, and deactivate the suppressor by preventing DNA binding and transactivation [26]. Moreover, mutation TP53 causes loss in tumor suppressor function, leaving the mutant protein capable of driving additional oncogenic processes, such as metastasis [27]. In this study, mutations in the KRAS exon 2 and TP53 exon 5 were analyzed. As expected, no frequent point mutations are detected in the analyzed genes in the Panc02 and BxPC3 cell lines (Appendix A). However, a silent SNP is found at codon 32 (TAT to TAC) in Panc02 cells. Since this is a synonymous SNP, it does not affect protein expression or function [28] (Appendix A).

To enable observation of whether KRAS affects alt-EJ, we generated Panc02 and BxPC3 cells expressing wild-type or oncogenic KRAS with the G12D mutation. To achieve this, we used previously designed wild-type and mutagenic KRAS plasmids cloned into the pLVX-DsRed-Monomer-C1 vector, and introduced them into cells. Since these cells are notoriously resistant to any kind of transfection reagents, we employed the lentiviral transduction system to express KRAS wild-type (KrasWT) and oncogenic KRAS^G12D^ (KrasMT). To estimate the transduction efficiency, we transduced the cells in parallel with a control virus carrying the fluorescent DsRed protein (pLVX-DsRed-Monomer-C1), followed by fluorescence imaging and FACS measurement. We achieve a transduction efficiency of almost 100% for Panc02 and BxPC3 cells (Appendix A).

To investigate the effect of oncogenic KRAS on alt-EJ components expression, we used established Panc02 and BxPC3 cell lines expressing KrasWT and KrasMT, and performed the immunoblot analysis. As shown in Figure 1A,B, exogenous expression of both KrasWT and the KrasMT clearly increases the expression level of Polθ, PARP1, Lig3, and Mre11, key factors of the alt-EJ pathway, in the mouse Panc02 cell line. As expected, the expression of the c-NHEJ components such as Ku80, Ku70, and Lig4 is not altered. In line, only the exogenous expression of oncogenic KRAS in the human BxPC3 cell line increases the expression levels of Polθ, PARP1, Lig3, and Mre11, while the KrasWT does not significantly increase the expression of alt-EJ components in the same human cell line. The expression of all c-NHEJ factors in BxPC3 cells remains unchanged. These data strongly support the theory that the expression of the oncogenic KRAS^G12D^ may result in enhanced expression of the alt-EJ pathway.

Intracellular protein expression is precisely regulated at the transcriptional and/or translational levels, and its deregulation can have deleterious consequences. KRAS, as a small GTPase transductor protein, transmits signals from extracellular receptors, mainly tyrosine kinase receptors, to the nucleus where it regulates the transcription of many proteins [29]. To address whether KRAS expression causes the transcriptional upregulation of DNA repair pathway components, in particular alt-EJ, we first conducted a microarray analysis of mouse Panc02 and human BxPC3 pancreatic cancer cell lines expressing either exogenous KrasWT or KrasMT. To our surprise, the overall gene set enrichment analysis (GSEA) of our microarrays does not identify any DNA repair pathways to be significantly enriched. In the additional targeted analysis of DNA double-strand break repair pathways, 10 genes of c-NHEJ and alt-EJ are upregulated in Panc02 cells. In BxPC3 cells, 9 genes of c-NHEJ and 10 genes of the alt-EJ pathway are upregulated, according to their signal-to-noise ratio. Although the GSEA analysis shows the upregulation of c-NHEJ and alt-EJ pathways in both Panc02 and BxPC3 cells, no significant enrichment is found (Figure 1C,D and Appendix A).

Next, we confirmed these results with qPCR analyzing mRNA expression of Polθ, PARP1, Lig3, and Mre11; the protein level of these is upregulated on the above immunoblots upon exogenous expression of the mutagenic KRAS (Appendix A). According to the microarray analysis, the mRNA level of the alt-EJ (Polθ, PARP1, Lig3, and Mre11) and c-NHEJ (Lig4, Ku80, and Ku70) core factors is also not altered. These data clearly indicate that KRAS does not regulate the expression level of alt-EJ components at the transcriptional level and, thus, a post-transcriptional mechanism must be involved.

### 3.2. KRAS Overexpression Promotes Proliferation and Arrest in S/G2-M Phase of the Cell Cycle in Mouse and Human Pancreatic Cancer Cells In Vitro

Cell proliferation is the process that results in an increase in cell number, and is defined by the balance between cell division and cell loss through cell death or differentiation. This process is also an important part of cancer development and progression. Many studies show that cancer cells are characterized by increased proliferation [30,31]. Given the well-known role of KRAS in cell proliferation, we investigated whether the activation of a point KRAS mutation results in a different proliferation rate compared to cells carrying the exogenous wild-type KRAS. For this purpose, we employed MTT assay on murine Panc02 and human BxPC3 pancreatic cancer cells expressing pLVX vector, KrasWT, or oncogenic KrasMT. We used untransduced cells as a control. Measurements were made after 24, 48, and 72 h. As shown in Figure 2A, murine Panc02 either bearing the exogenous oncogenic or wild-type KRAS shows increased proliferation compared to untransduced and pLVX transduced cells throughout the measurement. Interestingly, Panc02 cells harboring wild-type KRAS exhibit a higher proliferation rate than KRAS^G12D^-expressing Panc02 mainly after 24 and 48 h. On the other hand, human cell lines do not show the same trend of cell growth (Figure 2B). In this case, BxPC3 cells carrying the KRAS mutation also proliferate faster than pLVX and untransduced cells. However, the increase in the proliferation rate of BxPC3 KRAS wild-type cells is not higher than KRAS^G12D^-expressing BxPC3, as observed in the mouse cell line Panc02. Herein, cells harboring the KRAS mutation proliferate significantly faster than cells with KRAS wild-type throughout the measurement. Our results reveal that both the overexpression of KRAS wild-type and the activation of KRAS mutations can influence the proliferation rate in murine and human pancreatic cancer cell lines, resulting in an accelerated increase in cell number, which is a common characteristic of cancer.

Next, we investigated the impact of KRAS wild-type and oncogenic KRAS^G12D^ on cell cycle, as the cell cycle phase is one of the main determinants of DNA DSB repair pathway choice. In eukaryotic cells, DSBs can be repaired by three main mechanisms: canonical, non-homologous end-joining (c-NHEJ), homologous recombination (HR), and alternative end joining (alt-EJ). While c-NHEJ operates throughout the cell cycle, with predominant activity predominant in G1, HR and alt-EJ take place in the S and G2 phases of the cell cycle [10,12,32,33]. To gain insight into whether KRAS impacts the cell cycle, and thereby contributes to the choice of DNA DSB repair, Panc02 and BxPC3 cells either harboring wild-type or mutagenic KRAS^G12D^ were used and stained with propidium iodide (PI), followed by flow cytometric analysis. This approach allows for discriminating between cells in different phases of the cell cycle, based on their DNA content. pLVX-transduced Panc02 and BxPC3 cells were used as a control. The FACS analysis shows a statistically significant increase in the number of cells in the S/G2-M phase in both Panc02 and BxPC3 cells with the KRAS mutation (designated as KrasMT) compared to cells harboring KRAS wildtype (designated as KrasWT) and controls. Consequently, the increased number of Panc02 and BxPC3 cells with oncogenic KRAS in the S/G2-M phase is accompanied, to the same extent, by a decrease in the G1 phase. No significant changes in the proportion of cells in the G1 and S/G2M phases of the cell cycle are observed in either KRAS wild-type Panc02 and BxPC3 cells, or in the control cells (Figure 2C–F). However, Panc02 and BxPC3 cells expressing exogenous KrasWT show an increased tendency toward S/G2-M phase shift, although not significantly. Since the alt-EJ pathway predominantly operates in the S and G2 cell cycle phase, these results may provide a possible mechanism for how oncogenic KRAS^G12D^ induces an increase in alt-EJ activity in murine and human pancreatic cancer cell lines.

### 3.3. Exogenous KRAS^G12D^ Activates alt-EJ in Pancreatic Cancer Cells

As already shown above, in both murine and human pancreatic cancer cell lines, alt-EJ components are upregulated upon expression of the exogenous KrasG12D mutation. Based on the current understanding of cellular processes, we believe that upregulation of these key factors reflects increased alt-EJ biological activity. Moreover, the analysis of the cell cycle in these cell lines shows an increased number of cells in the S/G2-M phase, which supports the activity shift toward either the alt-EJ or HR pathway. Thus, we employed the Traffic Light Reporter (TLR) assay to measure and validate the mutagenic alt-EJ activity in Panc02 and BxPC3 cell lines. The TLR system, developed by Certo et al., enables simultaneous monitoring of homologous recombination (HR) and mutagenic activity of non-homologous end joining (mutNHEJ) in response to DNA damage in single cells [34]. Panc02 and BxPC3 cell lines expressing KRAS wild-type (KrasWT) or oncogenic KRAS^G12D^ (KrasMT) were transduced with a lentiviral vector containing the fluorescent TRL system, followed by flow cytometric analysis. mCherry-positive cells indicate a repair event induced by mutNHEJ, and eGFP-positive cells represent cells with the HR repair event. As expected, Panc02 and BxPC3 cells transduce with I-Scel alone produced only mCherry-positive cells, indicative of mutNHEJ at the reporter locus. On the other hand, both murine and human pancreatic cancer cells co-transduce with I-Scel, and donor templates produce either mCherry- or eGFP-positive cells (Figure 3 and Figure 4). Further analysis shows a higher HR capacity in control Panc02 and BxPC3 cells (Figure 3A and Figure 4A). BxPC3 KrasWT cells also exhibit an increasing fraction of events accounting for the HR pathway (Figure 4B). High mutNHEJ capacity is observed in both Panc02 and BxPC3 cell lines expressing the KRAS mutation (Figure 3C and Figure 4C). In line, the ratio of HR to mutNHEJ is lower in murine and human KRAS^G12D^-expressing cells compared to control (Figure 3E and Figure 4E). Moreover, increased mutNHEJ pathway events are also noted in KrasWT Panc02 cells (Figure 3B). However, the HR to mutNHEJ ratio in Panc02 cells between KRAS wild-type and the mutagenic KRAS is higher in Panc02 KrasWT cells, which is consistent with our results showing increased expression of alt-EJ proteins in these cells (Figure 1A and Figure 3E). Of note, as the amount of virus increases, we observe the expected dose-dependent increase in the total number of repair events in all analyzed Panc02 and BxPC3 variants (Figure 3D and Figure 4D). Taken together, these data clearly indicate that the oncogenic KrasG12D contributes to the activation of the alt-EJ pathway in both murine and human pancreatic cancer cells.

### 3.4. Polθ Ablation Affects Disease Progression in Animal Models of Pancreatic Ductal Adenocarcinoma

In pancreatic cancer research, several GEMM models were developed over the past two decades that provided new insights into its understanding [35]. Thus, to investigate in vivo the role of KrasG12D in the development of pancreatic intraepithelial neoplasia (PanINs), and their progression to pancreatic cancer, we used a KC mouse model. This pancreatic ductal adenocarcinoma mouse model is characterized by a pancreas-specific expression of the KrasG12D mutation, leading to the formation of PanINs and, ultimately, invasive PDAC that bears striking resemblance to tumor progression in humans [7]. We believe that in the presence of oncogenic KRAS, the alt-EJ pathway is a key player in tumorigenesis, and loss of alt-EJ components may prevent or delay PanIN lesions development and, ultimately, pancreatic cancer progression. Polymerase θ plays an essential role in alt-EJ, and its expression is generally repressed in healthy tissues, but significantly increased in cancers [19,22,36,37]. In addition, it is reported that Polq-null mice show no overt phenotype, despite the elevated genomic instability in erythroblasts [38]. Accordingly, due to the well-established function of Polθ in the alt-EJ pathway and tumorigenesis, we decided to breed the Polq knockout mice on a KC background (designated as qKC mice) to study the role of alt-EJ in the development of PanIN lesions and the transition to pancreatic cancer [20,39]. We observe that the pancreas of KC and qKC mice, especially in older animals, is larger than in control and Polq-deficient mice (designated as qKO), showing focal nodular parenchyma or pancreatic cancer (Figure 5A). Further histopathological analysis of mouse pancreases reveals significant differences in the formation of PanIN lesions between KC and qKC mice (Figure 5B). While 3 month old KC mice already show 56% PanIN lesions, the same age qKC mice have 40% of PanINs. This tendency continues with the age of the mice; only in 9 month old mice are no significant changes observed. Interestingly, in both the KC and qKC mice, one 4.5 month old mouse does not show any PanIN lesions (Figure 5C). We also noticed that the KC mice also have an increased presence of high-grade PanINs compared to the qKC mice (Figure 5D).

PanINs are the most frequent and well-known PDAC precursors. These neoplastic lesions are characterized by the conversion of the duct epithelial cells to a columnar phenotype with mucin accumulation [7,8]. PanINs are histologically subdivided into low-grade PanIN-1A and B, PanIN2, and high-grade PanIN-3, referred to as carcinoma in situ [2,40]. Moreover, tumor differentiation is separated into three stages, from grade 1 well-differentiated (G1) to grade 3 poorly differentiated (G3) tumors. The mucin content decreases with decreasing differentiation status. To visualize PanIN changes in the KC and qKC mouse models, we performed an alcian blue stain. Our results reveal strong alcian blue staining (acidic mucin stain) in both KC and qKC mice at 3, 4.5, 6, 9, and 12 months of age, both resembling grade 1 tumors (Figure 5E–G). However, pancreatic tissue of KC mice at 3 and 6 months old shows significantly more stained mucin-containing PanIN-like lesions with alcian blue. Furthermore, significantly less cytoplasm is observed in KC mice at 3, 4.5, and 6 months of age compared to qKC mice of the same age. Due to the very low number of PanIN foci in 1 month old KC and qKC mice, there is little or no blue staining. As expected, less mucin-rich PanIN lesions and more cytoplasm are observed in qKC mice compared to KC mice, indicating a delay in tumor progression due to polymerase θ deletion. Altogether, these results clearly support the role of polymerase theta in the development and differentiation of PanIN lesions in the mouse pancreas.

### 3.5. Oncogenic KRAS^G12D^ Promotes Expression of NHEJ Proteins in Pancreatic Ductal Adenocarcinoma

According to the current understanding, pancreatic cancer develops through the continuous progression of PanIN lesions into neoplastic transformation. Many studies show that low-grade PanINs already harbor KRAS mutations in more than 90% of the lesions, which are considered to be site-directed mutations that can cause sequential inactivation of suppressor genes. Consequently, this leads to genomic instability, which is a hallmark of most cancers. Genome instability is also associated with error-prone DNA double-strand break repair [12]. Assuming that the expression of alt-EJ components correlates with the activity of the mutagenic pathway, we would expect increased expression of alt-EJ factors in the presence of oncogenic KRAS. For this purpose, we performed immunohistochemical (IHC) analysis for alt-EJ and cNHEJ core components in KC and qKC mice (Figure 6A,B). We find strong nuclear expression of Polθ, PARP1, and Mre11, all alt-EJ key factors, in PanIN lesions, acinar cells (acini), and islets of KC pancreases in 3 and 6 month old animals. Ku70, a c-NHEJ factor, is also expressed in the same pancreas of KC mice. Interestingly, high expression of PARP1, Mre11, and Ku70 is also noted in 3 and 6 month old qKC mice, with PARP1 being the highest. Additionally, we observe visible Ku70 immunoreactivity in the cytoplasm of low-grade PanINs in 3 month old qKC mice. Positive immunohistochemical staining of Ku70 is also seen in islets and acini of control and qKO mice. In contrast, no expression of Polθ and Mre11 is detected in the ducts, acinar cells, or islets of 3 month old pancreases of control or qKO animals. Negative staining for PARP1 is also found in control mice. To our surprise, nuclear expression of PARP1 is noted in islets of qKO mice. Overall, we noticed that the development of PanIN lesions in qKC mice is similar to KC mice, accompanied by nuclear expression of the c-NHEJ and alt-EJ factors except for Polθ, which confirms the specificity of the antibodies. Our results show that the expression of alt-EJ components correlates with the development of precursor lesions of pancreatic ductal adenocarcinoma in the presence of oncogenic KRAS, and functions independently of the c-NHEJ pathway activity.

Next, we performed the immunohistochemical staining in normal human pancreas and pancreatic ductal adenocarcinoma (PDAC), to investigate the expression level of the above-mentioned alt-EJ and c-NHEJ components (Figure 6C,D). According to the mouse IHC analysis, we observe high expression levels of Polθ, PARP1, Mre11, and Ku70 in human PDAC tissue. In addition, positive IHC staining for Polθ, PARP1, and Mre11 is not detected in the ducts, acini, or islets of a normal pancreas. As expected, visible Ku70 immunoreactivity is found in the same normal pancreatic tissue. These findings clearly indicate the pathological relevance of alt-EJ in human pancreatic cancer.

### 3.6. Polθ Is Essential for PDAC Progression and Overall Survival

As the knockout of Polθ in the KC background delays disease progression, we also evaluated the effect of loss of Polθ on overall survival. Previous studies report that the survival rate of KC animals is higher than 450 days, and deaths of individuals begin after approximately 150 days [8]. Accordingly, in our research, we monitored the KC and qKC animals for 440 days. Survival studies conducted to explore the influence of Polθ deficiency on the oncogenic KrasG12D-induced pancreatic cancer mouse model show that KC mice begin to die after only 96 days of observation. In contrast, qKC mice have a longer lifespan, and the first deaths occurred only after 274 days (Figure 7A). Despite the prolonged survival of qKC mice, histopathological analysis reveals that nearly 65% of animals (n = 9/14) develop clinically significant pancreatic tumors. Additionally, 30% of qKC mice with PDAC have liver (n = 4/9) and lung metastases (n = 2/9) (Figure 7C–H; Appendix A). Interestingly, two males also have abdominal distention (Figure 7D). On the other hand, only 30% of KC mice (n = 6/20) exhibit PDAC (Figure 7B), and only one animal shows liver metastasis. Our results also show no correlation between gender and cancer progression in either the KC or qKC mouse model (Appendix A). Further microscopic examination of KC and qKC tumors shows rare histologic variants of pancreatic ductal adenocarcinoma and extra-pancreatic pathologies, such as sarcomatoid, fatty infiltration (FI), serous microcystic adenoma, and also pancreatobiliary type IPMN, which is present in only one KC mouse (data not shown). Sarcomatoid is observed in most qKC mice with PDAC (Appendix A). These findings suggest that, despite delayed tumor progression and longer animal survival associated with loss of polymerase theta, most qKC mice eventually develop PDAC that can metastasize to other organs.

Next, we decided to check whether the absence of Polθ also has an impact on clinical research. For this purpose, we used The Cancer Genome Atlas (TCGA) database of pancreatic adenocarcinoma patients with low and high expression of polymerase theta carrying KRAS wild-type or oncogenic KRAS^G12D^ to generate a survival curve. According to the mouse survival curve, patients harboring KRAS mutations and low expression of POLQ have a half longer median survival compared to patients with higher expression of POLQ. On the other hand, patients with KRAS wild-type tumor have a longer survival rate compared to the KRAS mutation patients, regardless of the POLQ expression level. Additionally, in the group of wild-type KRAS patients, individuals with lower POLQ expression show long-term survival (Figure 7I). These data clearly show that low expression of polymerase Q correlates with a higher rate of survival, and supports our data obtained with the PDAC mouse model.

### 3.7. Effect of Polθ Deficiency on Signaling Pathways in Oncogenic KRAS-Driven Mouse Models

The oncogenic KRAS activates different intracellular pathways such as PI3K, MAPK, or RAL-GEF to promote various cellular processes including proliferation, transformation, and survival [9,41]. To explore the effect of Polθ deficiency on cell signaling in murine KRAS models, we used immunohistochemistry for Ki67, proliferating cell nuclear antigen (PCNA), cyclin D1, cyclooxygenase-2 (COX-2), and the extracellular signal-regulated kinases (ERK1/2), which are the major components of the greater MAPK cascade that transduce growth factor signaling in the cell membrane. Ki67 and PCNA are proteins expressed in perinuclear or internal nuclear regions in all cell cycle phases except G0, making them excellent cellular markers for assessing cell proliferation in various tumors [42,43,44,45]. On the other hand, cyclin D1 is a proto-oncogene that acts as a cell cycle regulator that controls the transition from G1 to S phase in normal tissues. Its accumulation and mutations alter cell cycle progression, leading to increased cell proliferation and resulting in tumorigenesis [45]. In addition, cyclin D1 is also involved in the regulation of cell migration and invasion [46]. Immunohistochemical analysis shows a small amount of Ki67- and PCNA-labeled proliferative cells in the pancreas of the control and qKO mice, while in KC and qKC mice, the number of Ki67- and PCNA-positive cells is significantly elevated (Figure 8A,B). However, the observed increased Ki67 positivity is more pronounced in 9 month old KC pancreases compared to qKC pancreases at the same age. In contrast, no expression of cyclin D1 is detected in the control and qKO pancreases, but visible nuclear staining is observed in the pancreases of KC and qKC mice. Similar to immunohistochemical labeling of Ki67 and PCNA, cyclin D1 expression is higher in the pancreases of KC mice than in qKC mice (Figure 8A,B). Similarly to cyclin D1, ERK1/2 is considered a proto-oncogene that drives tumor cell proliferation, epithelial–mesenchymal transition (EMT), migration, and invasion [47]. Its activation is reported in several tumors. Despite the established role of ERK in driving cell cycle progression, it is also associated with other cellular events, such as senescence, autophagy, and apoptosis [47,48]. The phosphorylation of the ERK1/2 protein (p-ERK) as a mutant KRAS-activated signal is well-detected immunohistochemically. As shown in Figure 8, nuclear and cytoplasmic localization of p-ERK are observed in the islets of control, qKO, KC, and qKC pancreases. PanIN lesions are strongly stained in all KC and qKC animals. Additionally, the expression of p-ERK is noted in stromal cells of younger and older KC and qKC mice. Although a higher level of p-ERK expression is observed in the pancreatic tissue of KC mice compared to qKC, it is not statistically significant (Figure 8A,B). IHC analysis does not show positive staining for this protein in acinar cells of control, qKO, KC, or qKC animals. In the end, we evaluated the impact of Polθ absence on the inflammatory response. For this purpose, we used a component of the prostaglandin pathway, COX-2, whose synthesis can be upregulated by several cytokines, growth factors, and tumor promoters. In addition to its proinflammatory effects, the upregulation of COX-2 is noted in many types of cancer, indicating its role in carcinogenesis [49,50]. Immunohistochemical analysis for COX-2 demonstrates elevated expression in the cytoplasm of PanINs in KC and qKC mice, but not in the ductal cells, acini, or islets of control and qKO animals. No differences are observed in the expression of this protein in either KC or qKC younger animals. Surprisingly, significantly increased expression levels of COX-2 are observed in the pancreases of 9 month old qKC mice, but not in the pancreases of KC mice.

In conclusion, histochemical analysis for Ki67, PCNA, cyclin D1, p-ERK, and COX-2 reveals increased expression of these proteins in both aged KC and qKC mice. In addition, it is noted that, except for PCNA and COX-2, the rest of the analyzed proteins involved in KRAS-activated pathways show significantly higher expression levels in KC mice compared to qKC mice. These reports demonstrate that a deficiency of polymerase theta may have an inhibitory effect on pancreatic cancer progression in the early stages.

## 4. Discussion

In this study, we aimed to shed some light on the current knowledge of the role of Polθ in pancreatic cancer. In vitro, we evaluated the impact of oncogenic KRAS^G12D^ on the activity of alt-EJ in a panel of murine and human pancreatic cancer cell lines. We determined the effect of KRAS status on the proliferation rate and cell cycle profile of these cells, and, finally, examined which DNA repair mechanism predominates in the analyzed pancreatic cancer cells. In vivo, we focused on estimating the effect of Polθ deletion on the development of pancreatic intraepithelial neoplasia (PanINs) and their malignant transformation into pancreatic cancer in genetically engineered mouse models (GEMMs). Additionally, we investigated whether the absence of Polθ affects the activity of the other two DSB DNA repair mechanisms, HR and c-NHEJ, in selected transgenic mouse models and human pancreatic cancer.

### 4.1. The alt-EJ Pathway Proteins Are Upregulated in Pancreatic Cancer Cells Expressing Oncogenic KRAS^G12D^

In epithelial tumors, KRAS mutations are already detected in early pre-neoplastic lesions, suggesting a role of oncogenic KRAS in initiating cell transformation. However, further sequential genetic events, such as the inactivation of tumor suppressor genes, are required to ultimately lead to tumorigenesis [51]. Oncogenic KRAS may deregulate double-strand break (DSB) repair, resulting in the accumulation of pathological genomic DNA changes [52,53]. There are two main DSB repair mechanisms in higher eukaryotes, homologous recombination (HR) and non-homologous end joining (NHEJ), further subdivided into canonical NHEJ (c-NHEJ) and alternative end joining (alt-EJ) [10,32,33]. HR is considered an error-free repair pathway that uses a homologous sister chromatid as a template to faithfully repair the DSB [11,54]. The other main repair pathway is the error-prone c-NHEJ, which seals the two broken DNA ends with little or no sequence homology, frequently causing the appearance of small indels or chromosomal translocations [55]. Finally, alt-EJ is a mutagenic mechanism that uses microhomologies flanking the DNA ends, always resulting in large deletions and other sequence alterations at the repair junctions [11,12].

Interestingly, studies in several KRAS-mutated leukemic cell lines and primary T-ALL cells show that the activation of mutagenic KRAS is associated with increased expression of alt-EJ proteins [56]. Hähnel and colleagues observed enhanced expression levels of Lig3, PARP1, and XRCC1. In contrast, Ku70, Ku86, and Lig4, components of the c-NHEJ, are not altered in KRAS-mutation-expressing cells. Increased activity of alt-EJ proteins in DSB repair is also demonstrated in BCR–ABL-positive chronic myeloid leukemia cells [57]. These findings prompted us to investigate DNA repair pathways in murine and human pancreatic cancer cell lines with the KRAS G12D point mutation, the most common KRAS mutation in PDAC. Performed immunoblot analysis shows that the expression of oncogenic KRAS is associated with the increased expression of the alt-EJ components. Unexpectedly, a mouse cell line bearing exogenous wild-type KRAS also shows increased expression of the alt-EJ factors, which may indicate species-specific properties, or/and increased activity of KRAS. This is supported by the occurrence of KRAS amplification in various human tumors, leading to increased activity of this oncoprotein [58]. Further, recent studies by Wong et al. reveal that wild-type KRAS amplification is associated with enhanced Kras protein expression and poor survival in gastric cancer [59]. Thus, it is very likely that increased amounts of normal proto-oncogene proteins may alter the basic regulatory controls of cell proliferation, which is consistent with our results that demonstrate an increased proliferation rate in murine Panc02 cells harboring KRAS wild-type compared to control and KRAS-mutation-expressing cells. However, for *RAS* genes, low levels of a mutated protein appear to confer more malignant properties than the combined effects of multiple copies of the normal proto-oncogene [58,60,61], which may explain why RAS genes are more frequently activated by point mutations than by gene amplification.

Interestingly, we do not observe upregulation of the core components of the alt-EJ at the transcriptional level. It is widely assumed that changes in specific mRNA levels are always accompanied by commensurate changes in the encoded proteins and vice versa [62]. Thus, we postulate post-transcriptional regulation of expression such as RNA-binding proteins or non-coding RNAs e.g., miRNAs or lncRNAs [63,64]. In addition, comparative studies show that correlations between mRNA and protein levels in different model organisms can be relatively weak and uncertain or moderately positive, and that they also could vary between both experiments and organisms [65].

We further investigated DNA repair pathways in a pancreatic ductal adenocarcinoma mouse model, KC, and human PDAC. The immunohistological analysis also reveals high expression levels of Polθ, PARP1, and Mre11, as a result of KRAS mutagenic effects, confirming the correlation of alt-EJ components with activating KRAS mutation G12D. Ku70 is also expressed in mouse KC and human PDAC, but to a lesser extent compared with alt-EJ elements. Of note, elevated levels of Ku70 expression are also observed in low- and high-grade human bladder cancer [66]. These findings are consistent with the assumptions that the alt-EJ mechanism is highly regulated and functions independently, even when c-NHEJ is available [67,68,69].

Taken together, our results show that the expression of oncogenic KRAS contributes to the activation of the alt-EJ pathway only at a post-transcriptional level in pancreatic cancer cells, which points to the need for a broader investigation of the role of factors and processes occurring between transcription and translation in these malignancies.

### 4.2. Repair of DNA Double-Strand Breaks by alt-EJ Pathway in Pancreatic Cancer Cells Harboring Oncogenic KRAS^G12D^

DNA double-strand breaks (DSBs) are considered the most harmful, as a single unrepaired break may induce apoptosis, and a single incorrectly repaired DSB can lead to genomic instability and tumorigenesis [70]. Accordingly, to ensure genome integrity and cell homeostasis, cells evolved sophisticated DNA DSB repair mechanisms such as HR and NHEJ, including alt-EJ [10,32,33].

The fact that KRAS plays an important role in the regulation of the cell cycle, and c-NHEJ is active throughout the cell cycle with highest activity in G1, whereas HR and alt-EJ predominate in the S and G2 phases of the cell cycle [10,32,71], prompted us to formulate the hypothesis that oncogenic KRAS can activate the alt-EJ. The cell cycle analysis of Panc02 and BxPC3 cells reveals an increased number of cells in the S/G2-M phase in murine and human cells expressing oncogenic KRAS^G12D^, which may indicate the activity of either of the two DBS repair mechanisms, HR or alt-EJ. The activity of the HR and alt-EJ pathways in the S and G2 phase is compatible with previous work suggesting that mutagenic alt-EJ pathways may share the resection stage with HR [72]. However, given that alt-EJ is an error-prone mechanism, and its presence is often associated with genomic instability [12,73,74], we assumed that alt-EJ may outweigh the HR DSB repair in the presence of oncogenic KRAS. To address this question, we employed the Traffic Light Reporter (TLR) assay to measure the mutagenic alt-EJ repair activity in murine and human pancreatic cancer cell lines. We found that in both cell lines expressing oncogenic KRAS, alt-EJ is selected as the main repair pathway. Congruent with above results, increased mutNHEJ pathway events are also noted in KRAS wild-type Panc02 cells. These observations are consistent with our results showing an increased level of alt-EJ proteins in these cells. On the other hand, several studies show that defective DNA repair by HR results in the accumulation of chromatid breaks, and cells that cannot repair chromatid breaks by HR become more dependent on other alternative repair pathways [75,76,77]. As expected, high HR capacity is observed in the control Panc02 and BxPC3 cells, and BxPC3 expressing KRAS wild-type, which may indicate the absence of pathological changes in these cells through the accurate repair of DSBs [78]. 

In conclusion, our findings clearly indicate that the expression of oncogenic KRAS shifts the balance of DSB repair towards the highly error-prone alt-EJ pathway, and highlights the mutagenic properties of alt-EJ, making it the preferred DNA repair pathway in pancreatic cancer.

### 4.3. Depletion of Polymerase Theta Delays Pancreatic Cancer Progression in KrasG12D-Driven Mouse Model

DNA polymerase θ is a key component of the alt-EJ pathway [11,79]. Expression of Polθ is generally repressed in normal tissue, but is upregulated in several types of cancer [19,22,36]. In addition, high levels of Polθ are associated with poor prognosis and shorter relapse-free survival of patients with breast and lung cancers [36,37]. In contrast, molecular studies in mammalian cells demonstrate that the knockout of Polθ suppresses the alt-EJ pathway [79,80]. These reports led us to examine the effect of polymerase theta deletion in a well-established KrasG12D-driven mouse model of pancreatic cancer, known as KC. Our study is the first attempt to demonstrate the role of Polθ in PDAC progression using Polq knockout in mice with the KC background. We found that loss of Polθ results in slower tumorigenesis and PDAC progression. In line, fewer PanIN lesions are observed in qKC mice, which are especially visible in younger animals. A higher amount of low-grade PanINs is also noted in these mice. Accordingly, performed alcian blue staining reveals less mucin-rich PanIN lesions in qKC relative to KC mice, confirming the delayed cancer progression caused by polymerase theta deletion [81,82,83]. As expected, the overall survival of qKC mice is significantly increased compared to KC mice, highlighting the critical role of polymerase θ in pancreatic cancer progression. This work is supported by performed survival analysis of TCGA PDAC patients, which shows that low POLQ expression correlates with higher survival rates, regardless of KRAS status, compared to high POLQ expression in PDAC patients. Our findings are in agreement with the study by Shima et al. that demonstrates the increased survival of mice deficient in both ATM and polymerase theta in thymic lymphoma [38]. To our surprise, more qKC mice developed full-blown pancreatic tumors than KC mice, despite extended survival. In addition, qKC animals show progression to liver and lung metastases, as well as the presence of sarcomatoid, which can be found in KPC mice, a more aggressive KrasG12D-driven mouse model of PDAC. Moreover, abdominal distention is observed in two qKC males, which is also common in KPC mice [8]. These findings may suggest that polymerase theta deficiency may, on the one hand, prolong the survival of experimental animals, but, on the other hand, lead to the activation of other repair pathway factors or molecular processes, resulting in even more aggressive disease.

### 4.4. Polθ Deficiency Dampens Cell Proliferation, Migration, and Invasion in KrasG12D-Driven Mouse Model of PDAC

Given the established mechanistic role of KRAS in PDAC growth, we decided to explore the proliferation and invasion status under Polθ deficiency. Uncontrolled proliferation is one of the characteristic features of neoplasm [84]. One of the indexes of cell proliferation can be the Ki67 protein present in all phases, except G0 of the cell cycle, and found in proliferating cells [42]. In our study, Ki67 shows a statistically significant correlation between its expression and mice age. Ki67 expression is upregulated in older qKC mice that exhibit more low- and high-grade PanINs compared with younger animals. Experimental work from Klein et al. and Zinczuk et al. on human pancreatic tissues shows that enhanced Ki67 expression increases with the progressive stage of pancreatic intraepithelial neoplasia, implying intensified proliferation within the pancreatic duct epithelium [45,85]. Importantly, Ki67 expression is significantly lower in 9 month old qKC mice compared to KC mice of the same age, suggesting that Polθ deficiency may reduce the proliferation rate.

Another protein whose expression correlates with proliferation in pancreatic and other cancers is PCNA [43]. In our research, immunohistochemical analysis of PCNA performed on Polθ-deficient KC mice shows an increase in expression with animal age. Surprisingly, we do not observe reduced PCNA expression in qKC mice compared to KC mice, as was seen for Ki67 expression. In Zinczuk’s study [45] describing Ki67 and PCNA expression in pancreatic cancer, direct correlations between these proteins are demonstrated, revealing that an increase in the expression of one protein results in an increase in the expression of the other. In contrast, studies on breast cancer show that the expression of PCNA poorly correlates with Ki67 expression, suggesting that the usefulness of PCNA as a marker of proliferative activity appears to be limited [86,87]. Moreover, we observed PCNA expression in Polθ knockout mice compared to controls, indicating that depletion in polymerase θ may already itself affect molecular processes, independent of proliferation.

The negative regulator of the cell cycle for PCNA is cyclin D1, usually located in the cell nucleus, from which it disappears in the S phase. Overexpression of this protein is common in many cancers, and can be caused by chromosome translocation [88]. In our mouse model of PDAC lacking Polθ, we demonstrate that cyclin D1 expression increases with cancer progression, characterized by an increased presence of high-grade PanINs in older animals. The same trend is observed in KC mice, where the expression of this protein is significantly higher compared to qKC mice, indicating a higher degree of proliferation. Our findings are consistent with observations made in patients with pancreatic ductal adenocarcinoma, cysts, and pancreatitis, where a progressive increase in cyclin D1 expression is observed in correlation with PanIN staging [45]. Moreover, cyclin D1 overexpression shortens the transition time from the G1 to S phase, promoting cell progression and proliferation, which is one of the features of neoplastic transformation. The fact that cyclin D1 expression is induced by an activated RAS oncogene is evidence supporting the association of this protein with tumorigenesis [89].

Oncogenic KRAS is associated with increased phospho-extracellular signal-regulated kinase (ERK), which plays a crucial role in the proliferation, survival, and development of tumor cells [48]. Previous studies show that upregulation in phospho-ERK is associated with reduced survival in pancreatic cancer [90]. The present study examined the expression of phosphorylated ERK (p-ERK) as a hallmark of ERK activation in the KrasG12D-driven mouse model of PDAC lacking Polθ. Here, we demonstrate that increased expression of p-ERK enhances with age, consistent with PanIN progression in both qKC and KC mice. However, we do not observe the difference in p-ERK expression between the qKC and KC mouse model. These results suggest that the absence of Polθ might have no direct effect on the ERK signaling pathway whose activation promotes cancer–stromal interaction in PanIN cells due to oncogenic KRAS [91]. Moreover, qKO mice also show a slightly higher expression of p-ERK compared to control mice, suggesting that loss in Polθ may itself affect the MAPK/ERK pathway. Nevertheless, these mice do not develop cancer, and this effect is likely KRAS-independent.

Cyclooxygenase-2 (COX-2) is an important enzyme that synthesizes the proinflammatory mediators, prostaglandins, which play a key role in the generation of the inflammatory response and tumorigenesis [45,92,93,94,95,96,97,98]. Its expression is usually absent in most normal cells and tissues, but is highly induced in response to several cytokines, growth factors, and tumor promoters [99]. Our study shows a positive expression of COX-2 in both qKC and KC pancreases compared to healthy and qKO pancreases. Moreover, COX-2 expression in older animals with an increased number of high-grade PanIN lesions is significantly higher than that in young mice with a predominant number of low-grade PanINs in both qKC and KC. These results are compatible with the Maitra et al. study showing that the expression of COX-2 is significantly higher in high-grade PanIN lesions and poorly differentiated adenocarcinomas than in low-grade PanINs and moderately differentiated adenocarcinomas [50]. Overall, COX-2 expression in the pancreas increases with age and progression from normal ducts to low- and high-grade PanINs. Furthermore, we observed that COX-2 expression is significantly higher in older qKC mice than in KC mice. Several lines of evidence indicate that COX-2 is not only a critical player in tumor development, but also promotes dissemination of cancer cells to other organs [87,100,101,102,103]. Our finding demonstrates that a lack of Polθ can increase the expression of COX-2 in a mouse model of PDAC, which would explain the presence of an increased number of liver and lung metastases in the qKC mice.

## 5. Conclusions

In summary, our results support the oncogenic role of Polθ in development of pancreatic cancer, and provide evidence that the impairment of the alt-EJ pathway delays premalignant PanIN lesion development and pancreatic cancer progression. Moreover, a deficiency of Polθ in KC mice, despite prolonged survival, results in the eventual development of more aggressive full-blown PDAC with disseminated metastasis. Hence, we believe that a better understanding of Polθ, and in particular of the entire alt-EJ pathway, may prove an attractive target for pancreatic and other cancer treatment.

## Figures and Tables

**Figure 1 cancers-14-04077-f001:**
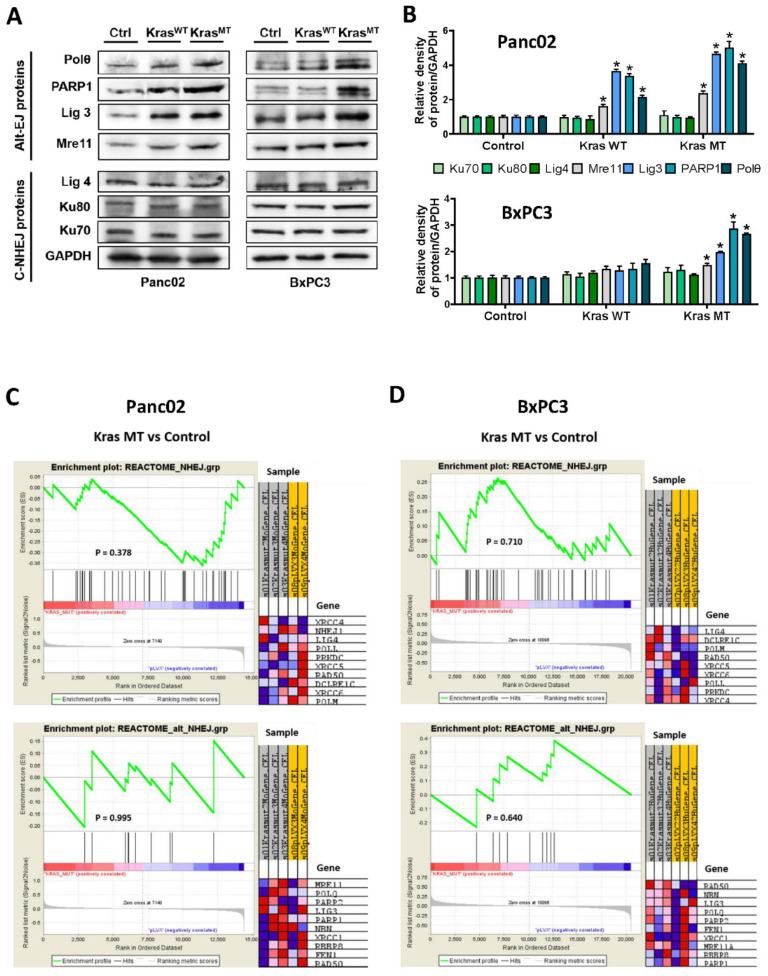
Impact of KrasWT and Kras^G12D^ mutation on expression levels of alt-EJ components. (**A**) Protein expression of Polθ, PARP1, Lig3, Mre11, Lig4, Ku80, and Ku70 in cell extracts isolated from transduced Panc02 and BxPC3 cells, and their respective quantification, are presented. Representative immunoblots from three independent experiments are shown. (**B**) Bar graphs show protein expression levels relative to GAPDH in Panc02 and BxPC3 determined by the means of densitometry of three independent experiments (mean ± SD; * *p* < 0.05 is considered as significant; Student’s *t*−test). The uncropped blots and molecular weight markers are shown in Appendix A. (**C**,**D**) Gene Set Enrichment Analysis plots for the Reactome NHEJ and alt-EJ in Panc02 and BxPC3 cells are shown. The heatmap on the right side of each panel visualizes the genes contributing the most to the enriched pathway. The green curve corresponds to the enrichment score (ES) curve, the running sum of the weighted enrichment score in GSEA. *p* values are reported within each graph (Panc02 control, n = 2; Panc02 KrasMT, n = 3; BxPC3 control, n = 3; BxPC3 KrasMT, n = 3).

**Figure 2 cancers-14-04077-f002:**
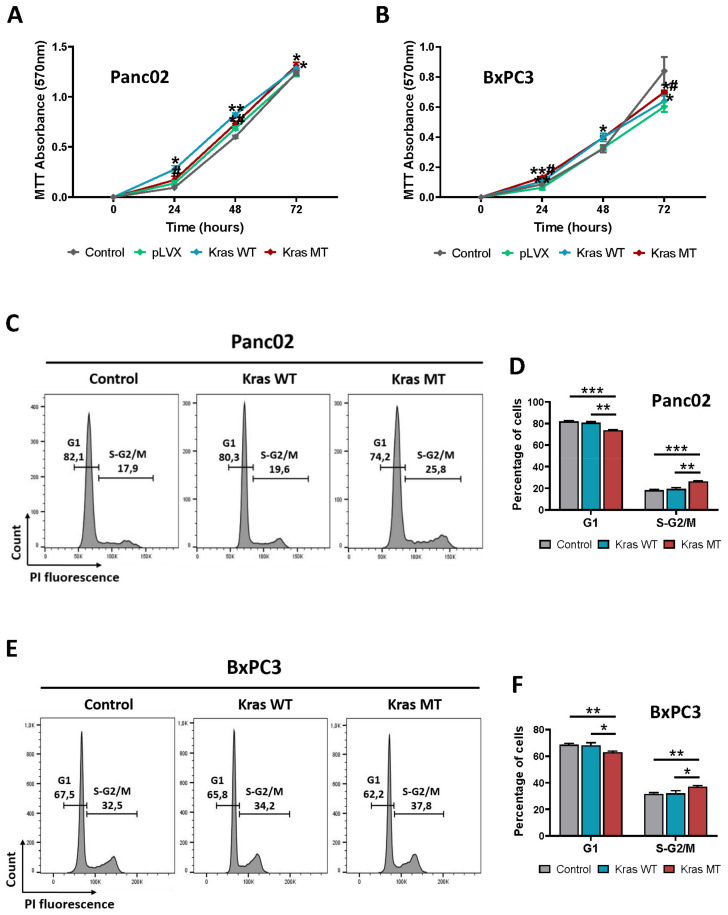
KRAS expression affects proliferation and alters cell cycle progression in pancreatic cancer cell lines. (**A**,**B**) The MTT assay was performed for 72 h and the absorbance of each well was read at 570 nm. A representative proliferation graph of (**A**) Panc02 and (**B**) BxPC3 cells from three independent experiments is shown (mean ± SD; ** *p* < 0.01, * *p* < 0.05 compared to pLVX cells; ^#^ *p* < 0.05 compared to KrasWT cells; multiple *t*-test). (**C**–**F**) Cell cycle analysis of (**C**) Panc02 and (**E**) BxPC3 cell lines with respective quantification is presented. Viable cells were collected by trypsinization, and DNA content was analyzed after PI staining. Representative flow cytometry histograms of cell cycle analysis from three independent experiments are shown. Quantification of data from (**D**) Panc02 and (**F**) BxPC3 cells is presented. Error bars represent mean ± SD; * *p* < 0.05, ** *p* < 0.01, *** *p* < 0.001 (Student’s *t*-test).

**Figure 3 cancers-14-04077-f003:**
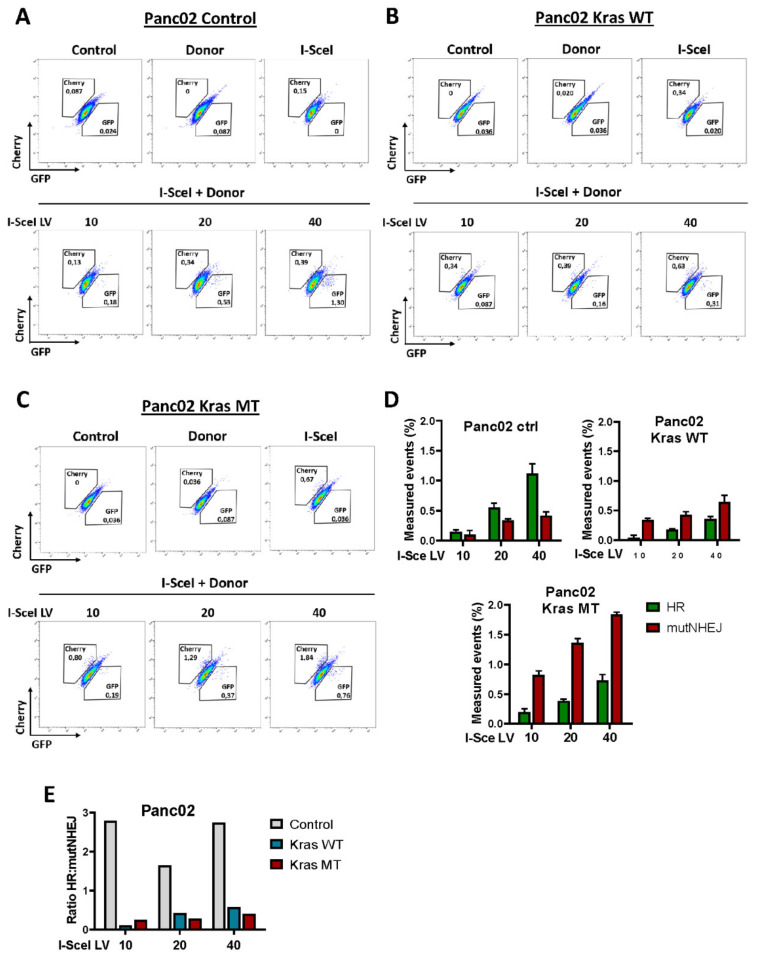
Traffic light reporter assessment of DNA repair fates in mouse pancreatic cancer cell lines. (**A**–**C**) Representative flow plots following transduction of Panc02 cells with donor and different doses of I-SceI lentivirus (LV) from three independent experiments are shown. Cherry-positive cells indicate a repair event induced by mutNHEJ, and GFP-positive cells represent cells with the HR repair event. (**D**) Quantification of data from all variants of Panc02 cells is presented. Error bars represent mean ± SD. (**E**) Ratio of HR to mutNHEJ based on data in panel D.

**Figure 4 cancers-14-04077-f004:**
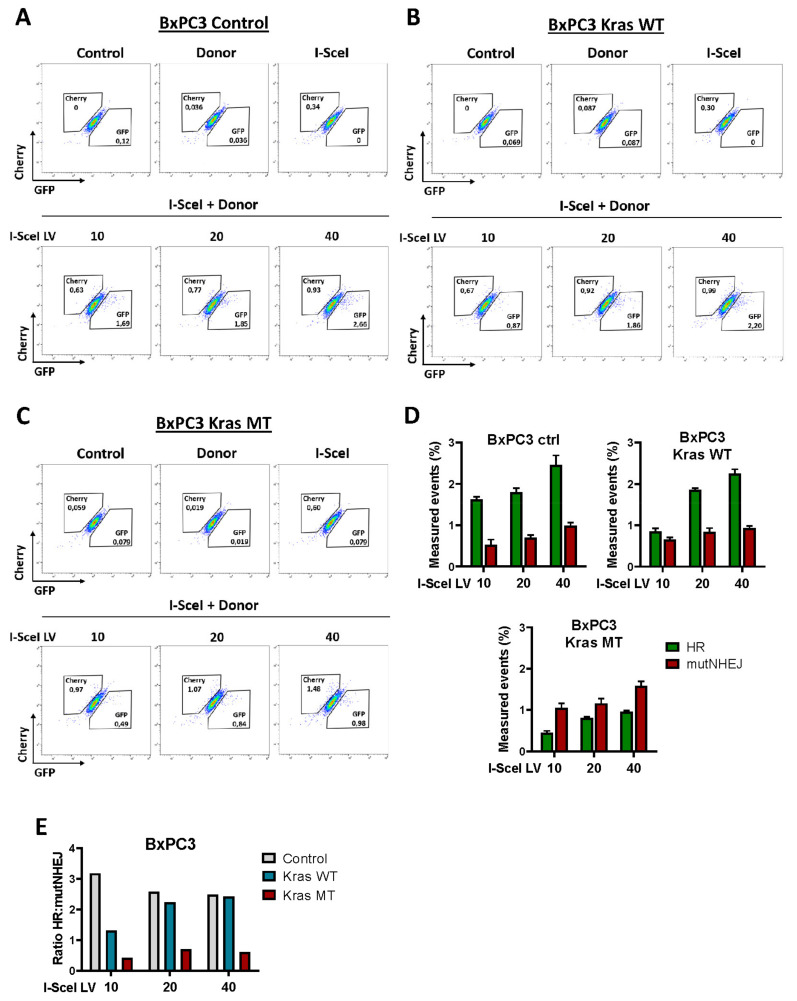
Traffic light reporter assessment of DNA repair fates in human pancreatic cancer cell lines. (**A**–**C**) Representative flow plots following transduction of BxPC3 cells with donor and different doses of I-SceI lentivirus (LV) from three independent experiments are shown. Cherry-positive cells indicate a repair event induced by mutNHEJ, and GFP-positive cells represent cells with the HR repair event. (**D**) Quantification of data from all variants of BxPC3 cells is presented. Error bars represent mean ± SD. (**E**) Ratio of HR to mutNHEJ based on data in panel D.

**Figure 5 cancers-14-04077-f005:**
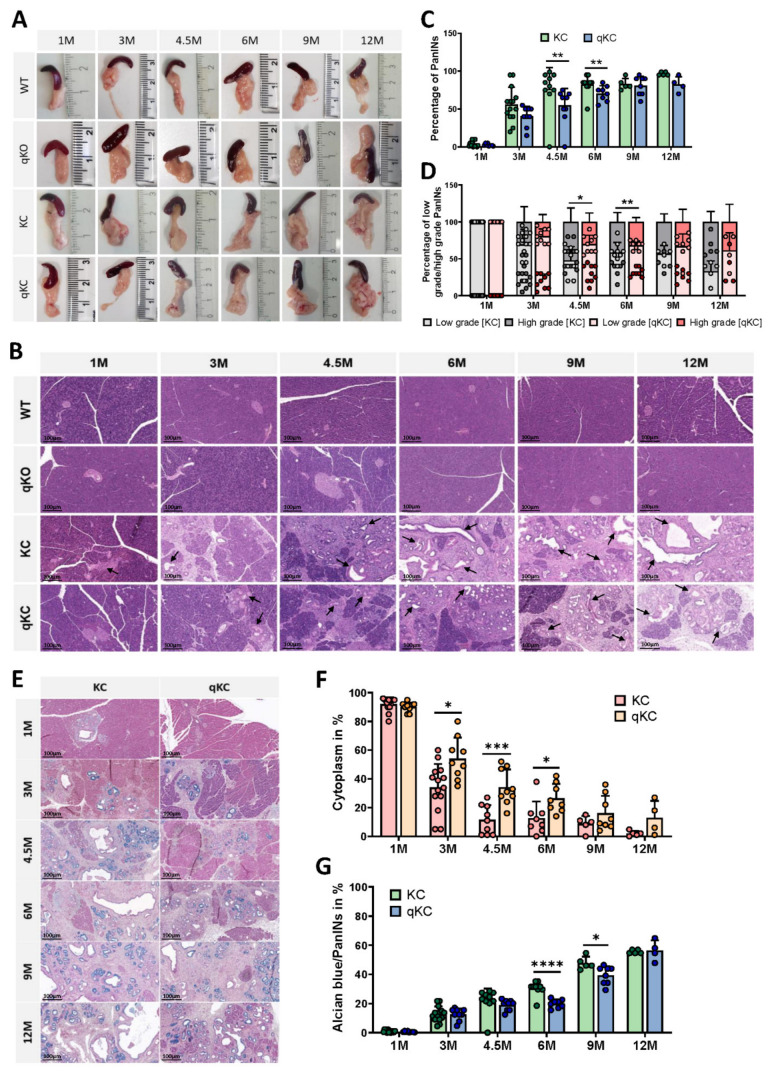
Deletion of POLQ affects tumor progression and differentiation in established pancreatic ductal adenocarcinoma in vivo. (**A**) Representative pancreases from 1 M, 3 M, 4.5 M, 6 M, 9 M, and 12 M WT, qKO, KC, and qKC animals. (**B**) Histology of the pancreas from age-matched wild-type, qKO, KC, and qKC mice at 1, 3, 4.5, 6, 9, and 12 months old. Arrows indicate PanIN lesions (hematoxylin and eosin stain; scale bars represent 100 µm). (**C**) Percentage content of PanIN lesions and (**D**) their histologic progression in KC and qKC mice at the age of 1 month (KC, n = 14; qKC, n = 9), 3 months (KC, n = 14; qKC, n = 9), 4.5 months (KC, n = 10; qKC, n = 10), 6 months (KC, n = 9; qKC, n = 8), 9 months (KC, n = 5; qKC, n = 8), and 12 months (KC, n = 5; qKC, n = 4). Error bars represent mean ± SD; * *p* < 0.05, ** *p* < 0.01, (Student’s *t*-test). (**E**–**G**) Alcian blue staining was performed to visualize differentiation. (**E**) Mucin-rich PanINs lesions are depicted in blue, cytoplasm in light red, and nuclei in dark red. Representative images of the pancreas from age-matched KC and qKC mice at 1, 3, 4.5, 6, 9, and 12 months old are shown. Scale bars represent 100 µm. Histological score of (**F**) cytoplasm and (**G**) PanINs in pancreas of KC and qKC animals at the age of 1 month (KC, n = 14; qKC, n = 9), 3 months (KC, n = 14; qKC, n = 9), 4.5 months (KC, n = 10; qKC, n = 10), 6 months (KC, n = 9; qKC, n = 8), 9 months (KC, n = 5; qKC, n = 8), and 12 months (KC, n = 5; qKC, n = 4). Error bars represent mean ± SD; * *p* < 0.05, ** *p* < 0.01, *** *p* = 0.0001, **** *p* < 0.0001 (Student’s *t*-test).

**Figure 6 cancers-14-04077-f006:**
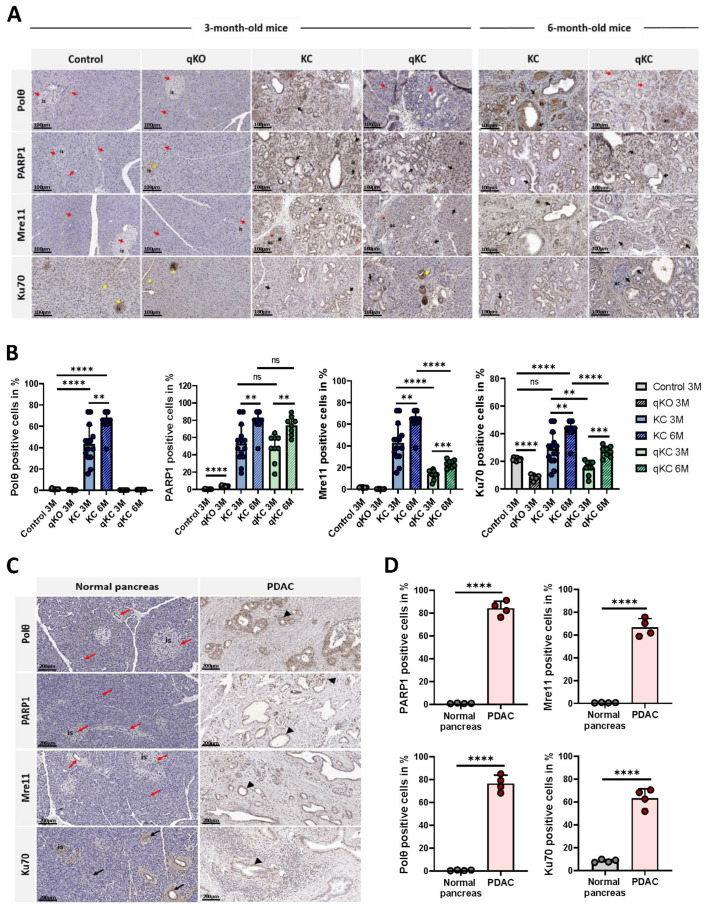
Impact of oncogenic Kras^G12D^ on the expression levels of major alt-EJ and c-NHEJ components in pancreatic ductal adenocarcinoma. (**A**) Representative immunohistochemical (IHC) staining images for Polθ, PARP1, Mre11, and Ku70 in the pancreas of control (WT), qKO, KC, and qKC mice at 3 months of age, and 6 month old KC and qKC mice. Nuclear expression of PARP1, Mre11, and Ku70 detected in PanINs, acini (ac) and islets (is) in 3 and 6 month old KC and qKC mice (black arrows). High expression of Ku70 seen in the cytoplasm of low-grade PanINs in 3 month old qKC mice (yellow arrow). Polθ expression is seen only in 3 and 6 month old KC mice (black arrows). Absence of PARP1 expression in normal ducts, acini, and islets (red arrows), but visible positive staining in islets of qKO mice (orange arrows). Ku70 nuclear and cytoplasmic staining detected in islets and acini of WT and qKO mice (yellow arrowheads). Absence of Polθ and Mre11 immunoreactivity in normal ducts, acini, and islets of WT and qKO mice (red arrows). Non-specific cytoplasmic staining for Polθ and Mre11 noted in KC and qKC mice (red asterisks). Scale bars represent 100 µm. (**B**) Histological score of Polθ-, PARP1-, Mre11-, and Ku70-positive cells in pancreas of experimental mice groups (control mice, n = 10; qKO mice, n = 10; KC mice of age 3 months, n = 14, 6 months, n = 9; qKC mice of age 3 months, n = 9, 6 months, n = 8). Error bars represent mean ± SD; * *p* < 0.05, ** *p* < 0.01, *** *p* = 0.0001, **** *p* < 0.0001; ns = not significant (Student’s *t*-test). (**C**) Representative immunohistochemical images for Polθ, PARP1, Mre11, and Ku70 in normal human pancreas and PDAC. Absence of Polθ, PARP1, and Mre11 immunoreactivity in normal ducts, acini, and islets (is) (red arrows). Positive immunohistochemical staining of Ku70 noted in normal ducts, acini, and islets (black arrows). High nuclear expression of Polθ, PARP1, Mre11, and Ku70 detected in PDAC (black arrowheads). Scale bars represent 200 µm. (**D**) Histological score of Polθ, PARP1, Mre11, and Ku70 positive cells in normal pancreas (n = 4) and PDAC (n = 4). Error bars represent mean ± SD; **** *p* < 0.0001 (Student’s *t*-test).

**Figure 7 cancers-14-04077-f007:**
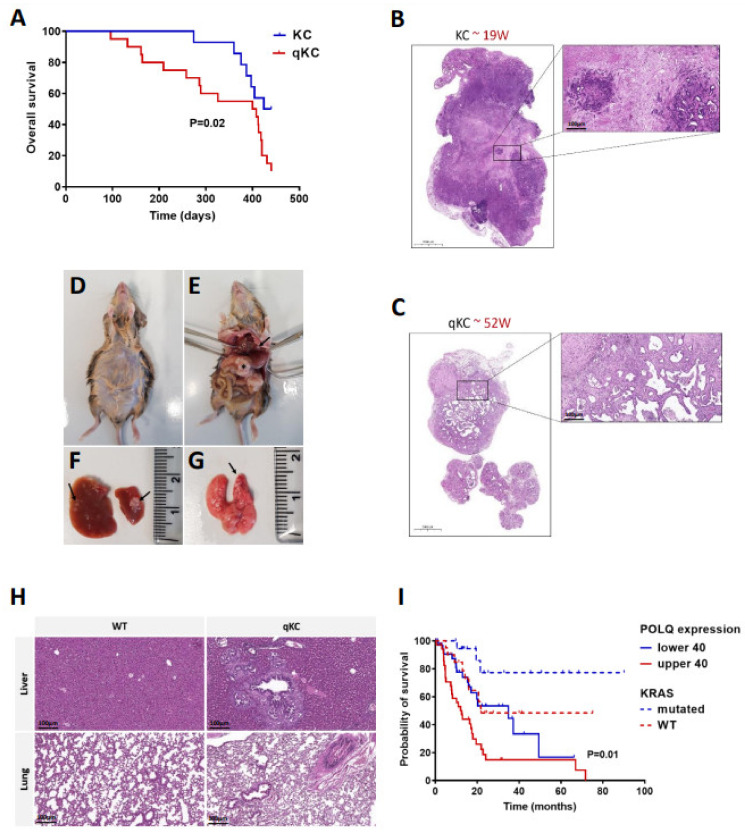
POLQ is significant for PDAC progression and overall survival. (**A**) Kaplan–Meier survival analysis for KC and qKC mice (KC mice, n = 20; qKC mice, n = 14; *p* = 0.02 by Mantel–Cox [log rank] test). (**B**) Representative images of pancreatic tumor from 19 week old KC mice (hematoxylin and eosin stain). (**C**) Representative images of pancreatic tumor from 52 week old qKC mice (hematoxylin and eosin stain; scale bars of enlarged images represent 100 µm). (**D**–**H**) Pathological photographs of metastatic PDAC in a representative qKC mouse. (**D**) Abdominal distention is noted, due to the accumulation of malignant ascites. (**E**) Primary PDAC in the pancreas (asterisk) and liver metastasis (black arrow). Tissue sample from the same mouse showing (**F**) multiple liver metastases and (**G**) lung metastasis marked with black arrowheads. (**H**) Histology of healthy liver and lung of wild-type mouse, and liver and lung metastases of qKC mouse. Scale bars represent 100 µm. (**I**) Kaplan–Meier survival analysis of TCGA PDA patients with low and high POLQ expression carrying KRAS wild-type or oncogenic KRAS (KRAS wild-type and lower POLQ expression, n = 20; KRAS wild-type and higher POLQ expression, n = 20; KRAS mutation and lower POLQ expression, n = 36; KRAS mutation and higher POLQ expression, n = 36; *p* value is indicated for comparison of patients with both KRAS mutation and higher or lower POLQ expression using the Mantel–Cox [log rank] test. Comparison of patient with both KRAS wild-type and higher or lower POLQ expression shows no statistical significance (*p* > 0.05).

**Figure 8 cancers-14-04077-f008:**
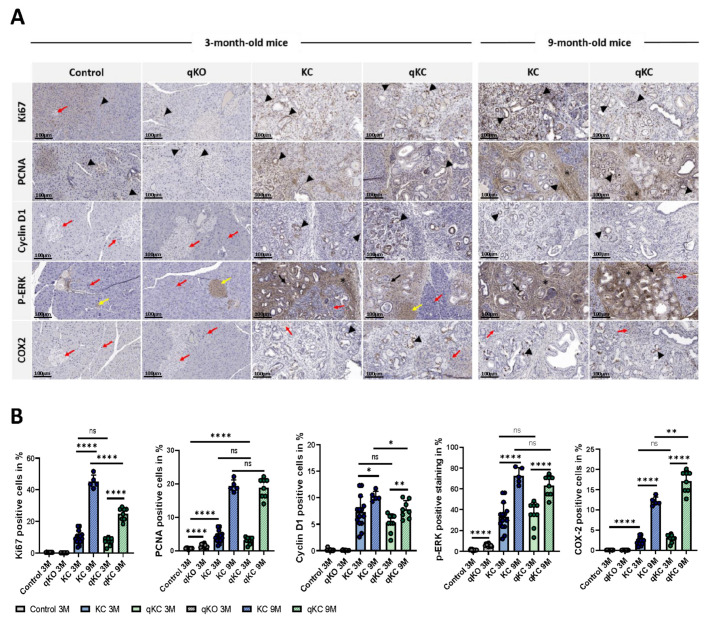
Signaling pathways in KC and qKC mice. (**A**) Representative immunohistochemical staining images for Ki67, PCNA, cyclinD1, p-ERK, and COX-2 in the pancreases of control, qKO, KC, and qKC mice at 3 months of age, and 9 month old KC and qKC mice. Strong nuclear staining of Ki67 detected in PanIN lesions of KC and qKC mice, and acinar cells of control, qKO, KC, and qKC animals (black arrowheads). No Ki67 expression in islets of control mice (red arrow). Positive staining of PCNA in acinar cells, islets, and ducts in all experimental animals (black arrowheads). Absence of cyclin D1 immunoreactivity in control and qKO mice (red arrows). Strong nuclear labeling of cyclin D1 in all KC and qKC pancreases (black arrowheads). P-ERK expression visible in islets of control, qKO, KC, and qKC animals (yellow arrows). In addition, strong positive staining of p-ERK noted in PanINs (black arrowheads) and stroma of KC and qKC mice (black asterisks). No expression of p-ERK in acinar cells of all experimental animals (red arrows). Cytoplasmic labeling of COX-2 detected only in PanIN lesions occurring in KC and qKC mice (black arrow). Absence of COX-2 immunoreactivity in control and qKO pancreases. Scale bars represent 100 µm. (**B**) Histological score of Ki67-, PCNA-, cyclin D1-, p-ERK-, and COX-2-positive cells in pancreas of experimental mice groups (control mice, n = 10; qKO mice, n = 10; KC mice of age 3 months, n = 14, 9 months, n = 5; qKC mice of age 3 months, n = 9, 9 months, n = 8). Error bars represent mean ± SD; Error bars represent mean ± SD; * *p* < 0.05, ** *p* < 0.01, *** *p* = 0.0001, **** *p* < 0.0001; ns = not significant (Student’s *t*-test).

## Data Availability

The data presented in this study are available in this article (and Appendix A).

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
