# Peer review of "DNA Polymerase Theta Plays a Critical Role in Pancreatic Cancer Development and Metastasis"

_cancers, 2022, doi:10.3390/cancers14174077_

Round 1

Reviewer 1 Report

In this report, the authors investigate the role of MMEJ in PDAC. They show that the G12D KRAS mutation increases levels of MMEJ proteins. Using an in vivo repair assay they also show that MMEJ repair is increased in KRAS G12D. The assay is sensitive to donor concentrations and tends bias towards HR at higher donor levels. However, the authors take this into account and show that even at higher donor levels mutant KRAS prefers MMEJ. Finally, they show that pol theta is responsible for increased proliferation in PDAC.

This reviewer believes that these findings should be of interest to readers of Cancers. These data shed light on DNA damage repair in PDACS. I have only minor comments.

Minor comments.

1.       Line 217: “Since this is a synonymous SNP, it does not affect protein expression and function 217 (Supplementary Figure S1A)”. How do the authors know this? Silent mutations have been known to affect protein expression, particularly translation. Unless it has been shown in the literature or the authors tested it, this cannot be concluded here.

2.       Figures should be enlarged. I don’t think this journal charges by page numbers so there is no need to make the figures so tiny. Take for example figure 1c,d. Is very hard to read. I would enlarge figures like figure 1 to take the whole page.

Author Response

  1. Line 217: “Since this is a synonymous SNP, it does not affect protein expression and function 217 (Supplementary Figure S1A)”. How do the authors know this? Silent mutations have been known to affect protein expression, particularly translation. Unless it has been shown in the literature or the authors tested it, this cannot be concluded here.

The SNP TAT and TAC code for the same amino acid and therefore the function of KRAS is expected to be unaffected. However, this SNP may indeed affect the transcription, translation and mRNA stability. We did not investigate this but we would not expect that expression level of endogenous KRAS would affect our results. Moreover, we overexpressed KRAS WT and M using a lentiviral expression system thus, the exogenous expression of KRAS proteins exceeds by far the level of endogenous protein. In addition, this SNP was detected in C57BL/6 WT mice and did not show any specific phenotype.

Genomic sequencing of key genes in mouse pancreatic cancer cells.

Wang Y, Zhang Y, Yang J, Ni X, Liu S, Li Z, Hodges SE, Fisher WE, Brunicardi FC, Gibbs RA, Gingras MC, Li M.

Curr Mol Med. 12, 331-41 (2012).

  1. Figures should be enlarged. I don’t think this journal charges by page numbers so there is no need to make the figures so tiny. Take for example figure 1 c,d. Is very hard to read. I would enlarge figures like figure 1 to take the whole page.

Changes have been made as requested.

Reviewer 2 Report

In this article, Smolinska et al. have investigated the role of alternative non-homologous end joining repair mechanism in pancreatic cancer. Although interesting, this article requires further experiments before it can be accepted. The specific comments are

Minor comments

1.      English can be improved

2.      Figures (1, 2, 3, 4) are difficult to see. I advise to increase image resolution and font

3.      Figures 3 and 4 should have the same scale bars in all graphs

Major comments

First part of the study is importantly less developed than second part ( in vivo work). I believe that additional experiments regarding this first part are needed. For example.

1.     Authors should use any other PDAC cell lines, as well as ´´normal´´ pancreatic cell lines (HPDE or HPNE) to see weather in cells with no artificial introduction of KRAS mutation they can see similar increase in alt-EJ proteins.

2.     Cell cycle, proliferation and TLR assay would need to be performed in cells that are not able to perform alt-EJ, for example cells where LIG1, LIG3 or Polθ is silent by siRNA, CRISPR or mutated. 

Author Response

Please find attached the point-by-point response to your questions as PDF file.

Reviewer 3 Report

1. The authors should also talk about other KRAS mutations such as G12C and G12R in the introduction and whether all KRAS mutations result in similar phenotypes as KRAS G12D.

2. The authors chose to over express KRAS WT and MT instead of generating a KRAS G12Dknock-In cell line or expressing the MT in an KRAS null cell system. Do the authors attribute KRAS overexpression to an increase in expression of alt-EJ marker proteins?

3. Does KRAS overexpression also lead to suppression of tumor suppressor genes (mRNA) and proteins (immunoblot)? Does KRAS overexpression lead to an increase in overall PCNA levels?  It will be imperative if the authors address this with some data.

4. Unless functional assays are performed, to claim that KRAS expression may lead to enhanced activity just based on increased expression of certain proteins may be considered as an over interpretation and the authors should revisit that statement on Line 239-240, page 6.

5. The current MTT assay (Figure 2) all by itself is not conclusive to predict the impact of KRAS overexpression on cellular proliferation and hence the authors need to compliment the MTT assay with other assays such as colony forming assay or atleast let the MTT assay run for 5 to 7 days. In addition, the authors could very well perform soft agar assay. Alternatively, the authors should clarify if increase in proliferation is an indirect outcome of reduced cell death.

6. To validate the claim that KRAS MT contributes to alt-EJ, the authors should test the hypothesis in presence of Pol theta and/or PARP inhibitors in the Traffic light reporter assay and measure if that blunts alt-EJ functionality.

7. The IHC data suggests that oncogenic KRAS not only correlates with expression of alt-EJ components but also with c-NHEJ, so the authors need to rephrase the result title “Oncogenic KRASG12D promotes alt-EJ activity in pancreatic ductal adenocarcinoma” to be more representative of the overall data Line 442.

8. Instead of PolQ KO, have the authors considered using PolQ inhibitor in their assays. Given that even PARP is over expressed due to oncogenic KRAS, will PARP inhibitor also suppress cellular proliferation and prolong survival?

9. Minor error – loss of PolQ instead of loss PolQ line 503

Author Response

(The authors gave the same response as above.)

Reviewer 4 Report

This is a study examining the role of alt-EJ repair in development of pancreas adenocarcinoma. There are several questions that I have.

11.      What was the reason for using KC versus KPC models?

22.       The tumor microenvironment (TME) plays are large role in PDAC development. Were there changes in the immune infiltrate within the TME when comparing the knockout (qKC) versus KC mouse models?

33.       Can the authors provide more data on the human PDAC samples that were stained? Did they undergo surgery, what pathologic stage, did they receive neoadjuvant chemo or chemoradiation therapy, etc…)?

44.       What was the cause of death for the KC versus qKC mice for the survival analysis? Was it cancer related or other since it appears many did not develop tumors?

Author Response

  1. What was the reason for using KC versus KPC models?

In our experiments, we did not use the KPC model because this model carries a p53 mutation in addition to the KRAS G12D mutation. In the literature, this is the second most common mutation in pancreatic cancer. KRAS mutation occurs in almost 100% of human pancreatic cancer, is already detectable in the very early premalignant lesions and is considered the driver mutation which may lead to stepwise accumulation of subsequent mutations, e.g. p53 that result in full-blown pancreatic cancer. In our hypothesis, we considered the occurrence of the KRAS MT as a key event that shifts the balance of the DNA repair mechanism toward alt-EJ, believed to carry out inaccurate DNA repair and causing thereby mutations in tumor genes. Second of all, using the KPC model would result in loss of Pol theta and two genes (KRAS and p53) mutations thus, we could not decipher whether the effect of Pol theta is attributed to KRAS MT or mutant p53.

  1. The tumor microenvironment (TME) plays are large role in PDAC development. Were there changes in the immune infiltrate within the TME when comparing the knockout (qKC) versus KC mouse models?

We did not investigate the TME as it would be beyond the scope of the study and would be a project for itself. However, we examined the expression of COX-2, an inflammatory protein that has been found as a key element mediating microenvironment changes. This protein is highly expressed in the tissues of diverse tumors and affects different aspects of tumor progression. Based on our studies, we observed increased expression of COX-2 in old qKC animals that also show a more aggressive phenotype of pancreatic cancer. Thus, we can assume that Pol theta knock-out gene may affect TME but further precise studies are needed.

Cyclooxygenase-2 in cancer: A review

Hashemi Goradel N, Najafi M, Salehi, Farbood B, Mortezaee K

J Cell Physiol. 234, 5683-5699 (2019)

COX-2 Signaling in the Tumor Microenvironment.

Zhang Y, Tighe S, Zhu YT.

Adv Exp Med Biol. 1277, 87-104 (2020).

  1. Can the authors provide more data on the human PDAC samples that were stained? Did they undergo surgery, what pathologic stage, did they receive neoadjuvant chemo or chemoradiation therapy, etc…)?

All pancreas tissue was collected from patients who underwent pancreas surgery due to PDAC. The tumor was confirmed by histological analysis through a pathologist in our Department of Pathology, University Medicine Greifswald. Healthy tissue was obtained from the healthy edge surrounding the tumor. The patients did not undergo any chemotherapeutic or radiation treatment before surgery. All tissues were collected according to the protocol set by the ethics committee.

We involved this information in the methods.

  1. What was the cause of death for the KC versus qKC mice for the survival analysis? Was it cancer related or other since it appears many did not develop tumors?

As published in the original papers by Hingorani et al. 2003 and 2005 KPC mice uniformly develop pancreatic cancer. However, the KC mice have significantly shorter life expectancy compared to WT controls though they only in some cases develop pancreatic cancer. This is not completely understood but all KC mice develop severe pancreatic premalignant lesions which in aging animals result in a loss of pancreatic structure. Consecutively, this may lead to complete/major loss of healthy tissue and in consequence to loss of pancreatic function and death of the animals. This must be investigated.

Trp53R172H and KrasG12D cooperate to promote chromosomalinstability and widely metastatic pancreatic ductal adenocarcinoma in mice

Hingorani SR, Wang L, Multani AS, Combs C, Deramaudt TB, Hruban RH, Rustgi AK, Chang S, Tuveson DA.Hingorani SR, Wang L, Multani AS, Combs C, Deramaudt TB, Hruban RH, Rustgi AK, Chang S, Tuveson DA.

Cancer Cell. 7, 469–483 (2005).

Preinvasive and invasive ductal pancreatic cancer and its early detection in the mouse
Hingorani SR, Petricoin EF, Maitra A, Rajapakse V, King C, Jacobetz MA, Ross S, Conrads TP, Veenstra TD, Hitt BA, Kawaguchi Y, Johann D, Liotta LA, Crawford HC, Putt ME, Jacks T, Wright CV, Hruban RH, Lowy AM, Tuveson DA.

Cancer Cell. 4, 437–450 (2003).

Round 2

Reviewer 2 Report

All my concernes have been adressed.

Reviewer 3 Report

The authors have sufficiently addressed my concerns.

Reviewer 4 Report

The authors have addresses my questions and paper can be accepted